# Proteomic Profiling Reveals Increased Glycolysis, Decreased Oxidoreductase Activity and Fatty Acid Degradation in Skin Derived Fibroblasts from LHON Patients Bearing m.G11778A

**DOI:** 10.3390/biom12111568

**Published:** 2022-10-26

**Authors:** Shun Yao, Xiaoli Zhang, Xiuxiu Jin, Mingzhu Yang, Ya Li, Lin Yang, Jin Xu, Bo Lei

**Affiliations:** 1Henan Provincial People’s Hospital, Zhengzhou 450003, China; 2Henan Eye Hospital, Henan Provincial People’s Hospital, Zhengzhou 450003, China; 3Academy of Medical Sciences, Zhengzhou University, Zhengzhou 450052, China

**Keywords:** LHON, fibroblast, proteomics, glycolysis, oxidoreduction, fatty acid metabolism

## Abstract

LHON is a common blinding inherited optic neuropathy caused by mutations in mitochondrial genes. In this study, by using skin fibroblasts derived from LHON patients with the most common m.G11778A mutation and healthy objects, we performed proteomic analysis to document changes in molecular proteins, signaling pathways and cellular activities. Furthermore, the results were confirmed by functional studies. A total of 860 differential expression proteins were identified, containing 624 upregulated and 236 downregulated proteins. Bioinformatics analysis revealed increased glycolysis in LHON fibroblasts. A glycolysis stress test showed that ECAR (extra-cellular acidification rate) values increased, indicating an enhanced level of glycolysis in LHON fibroblasts. Downregulated proteins were mainly enriched in oxidoreductase activity. Cellular experiments verified high levels of ROS in LHON fibroblasts, indicating the presence of oxidative damage. KEGG analysis also showed the metabolic disturbance of fatty acid in LHON cells. This study provided a proteomic profile of skin fibroblasts derived from LHON patients bearing m.G11778A. Increased levels of glycolysis, decreased oxidoreductase activity and fatty acid metabolism could represent the in-depth mechanisms of mitochondrial dysfunction mediated by the mutation. The results provided further evidence that LHON fibroblast could be an alternative model for investigating the devastating disease.

## 1. Introduction

With an incidence of about 1/31,000–1/54,000, LHON (Leber Hereditary Optic Neuropathy) is the most common hereditary optic neuropathy [1]. The average onset for patients is around 20 years old, with subacute vision loss in both eyes, which may lead to blindness and bring a heavy burden to individuals and their families [2,3,4]. Nevertheless, the detailed mechanism underlying the diseases is far beyond understanding.

About 30 mitochondrial gene variants contribute to the occurrence of LHON, among them, three mutations, including m.11778G>A (MT-ND4) [5], m.14484T>C (MT-ND6) [6], and m.3460G>A (MT-ND1) [7], are the most frequent. The proteins encoded by these genes are subunits of complex I of the respiratory chain. The mutated proteins directly cause mitochondrial respiratory dysfunction, ultimately resulting in LHON. Therefore, these mutations are regarded as primary mutations [8]. Although mtDNA mutations are currently known to be a major cause of LHON, not all individuals with pathogenic variants present the phenotype. In addition, multiple mitochondrial gene variants increase penetrance synergistically [9,10,11,12]. Some nuclear gene mutations combined with mtDNA variants also aggravate the phenotype of LHON [13,14]. The phenomenon that different mutation types and numbers contributed to different phenotypes reflects the heterogeneity and complexity of LHON.

Fibroblasts have been used recently as an alternative cell model for genetic disease research. Although fibroblasts are not the primary affected tissues, their pathology changes may represent those in the sensitive cells, especially the mitochondrial function [15]. Our previous study also showed that LHON patient-derived skin fibroblasts (m.G11778A) presented mitochondrial dysfunction and could be used as a succedaneous cell model [16]. 

Proteomics is an effective approach for exploring the underlying molecular changes. Tun A. et al. performed mitochondrial proteomic analysis using skin fibroblasts from LHON patients with the m.G11778A mutation and found that bioenergetic derangements and poor protein quality control systems lead to different penetrance [17]. The mitochondrion is an important organelle that regulates various cellular activities [18,19,20]. Mitochondrial dysfunction caused by genetic variation may lead to a variety of function disorders and even changes in protein expression profile, which may also contribute to heterogeneity. Although it may not reveal the pathogenesis of LHON comprehensively from the perspective of mitochondrial damage, it provides a robust technique to explore the diversity of protein profile and the changes in molecular activities of LHON. 

In the current study, we performed proteomic profiling on skin fibroblasts derived from LHON patients bearing m.G11778A to find the affected cellular activities. These changes may be the consequences of mitochondrial dysfunction or may act as secondary factors driving disease progression. The results were beneficial for revealing the pathogenesis and helpful for finding new therapeutic targets.

## 2. Materials and Methods

### 2.1. Subjects and Fibroblasts Construction

This study was conducted on the basis of the Declaration of Helsinki. Subjects included in this study were recruited from Henan Provincial People’s Hospital, Henan Eye Hospital. All participants were informed benefits and risks and signed the content forms. This study was approved by the Ethics Committee of Henan Eye Hospital (ethic number: HNEECKY-2019(12)). 

Men participants came from 4 families. Two controls were around 20 years old. Ophthalmological examinations showed the binocular visual acuity in both eyes was 1.0 or above, and no abnormality was found in a visual field examination. The patients LHON-1 and LHON-2 were 28 and 32 years old. The ophthalmic examinations showed pathological optic discs with incomplete boundaries, and the optic nerve has atrophy of varying degrees. Gene sequencing reported they both suffered mutation m.G11778A. Finally, they were diagnosed as LHON bearing m.G11778A. Detailed information can be found in a previous article [16].

A volume of 2 mm^3^ of skin tissue was excised from the participant’s arm and soaked in 75% alcohol for 2 min. Tissues were incubated with a complete medium (Lab050-NP, Kuisai, Shanghai, China) at 37 °C for 30–60 min after being shredded into pieces. The pieces were dispersed into flasks respectively and cultured with 2 ml complete medium for 2 h. Change the medium every 3 days until the scattered cells growing fuse into a sheet. Cells were trypsinized and re-cultured in a new flask. Follow-up experiments were performed after the cells had grown to a sufficient number.

### 2.2. Sample Preparation

Fibroblasts were used for proteomic analysis, and each group contained 3 replications. Cells were lysed in urea lysis buffer (8 M urea, 50 mM NH_4_HCO_3_, 1X protease inhibitor cocktail) at 25 °C for 5 min. Samples were fragmented by ultrasonic sonication and centrifuged (14,000× *g*) for 10 min at 20 °C. Supernatants were collected and quantified using BCA (Solarbio, Beijing, China). A total of 100 μg proteins of each sample was reduced by 10 mM DTT (Sigma-Aldrich, St. Louis, MO, USA) for 1 h at 37 °C and alkylated by 40 mM IAA (Sigma-Aldrich, St. Louis, MO, USA) for 1 h at RT temperature. Samples were loaded in pre-equilibrated ultrafiltration tubes (10 kDa) and centrifuged (14,000× *g*) twice for 15 min at 15 °C. Tubes were then washed 3 times with 50 mM NH_4_HCO_3_ by 14,000× *g* for 15 min. Samples were loaded in new tubes and incubated with 3 μg tyrisin for 13–16 h at 37 °C. Peptides were washed twice using H_2_O_2_ and centrifuged at 14,000× *g* for 20 min, 4 °C. A volume ratio of 1% formic acid was added to the collection tube to terminate the enzyme digestion and vacuum drying at 60 °C. Digested peptides were qualified using Nanodrop 2000 spectrophotometer (Thermo Scientific, Waltham, MA, USA) after resuspending in 0.1% formic acid.

### 2.3. LC-MS and MS Analysis

The liquid chromatography model used in this study was ekspert nanoLC 415 (AB SCIEX, Framingham, MA, USA), and the mass spectrometer was Triple TOF 6600 (AB SCIEX). The digested peptides (1.5 μg) were loaded into the Trap column of AB SCIEX (10 mm × 0.3 mm, C18 packing size was 5 μm, 120 A) with phase A (0.1% formic acid, 2% ACN, 97.9% water), and the flow rate was 10 μL/min. Different gradients of phase B (97.9% acetonitrile, 2% water, 0.1% formic acid, *v*/*v*/*v*) were used to elute the Trap column for 91 min at a flow rate of 400 nL/min. Then, peptides were separated on an analytical column (150 mm × 0.3 mm; C18: 3 μm, 120 A) to form a charged spray followed by MS detection. 

In MS analysis, protein identification and quantification were performed using a spectra library by DIA. The DDA spectrum parameters were as follows: TOF MS accumulation time was 0.25 s; mass scan range was 300–1500 *m/z*; ion charges were +2 to +5; ion exclusion time was 15 s; candidate ions per cycle was 60 and mass tolerance was 50 ppm. For DIA analysis, a variable window assay calculator (AB Sciex, version 1.1, Framingham, MA, USA) was used to optimize the 100 scanning windows. The accumulation time for MS1 was 50 ms, and 40 ms for MS2. The ion spray floating voltage was 2300 V.

### 2.4. Data Processing

The raw data collected by DDA was searched in the database using Spectronaut software (version 15). The UniProt Swiss human database (UniProt release 2020_11) was used as a library database. The DIA data processing was referenced previous study [21].

### 2.5. Bioinformatic Analysis

The original quantitative values were converted to log2 to satisfy the normal distribution, and the data were normalized using an R package (PreprocessCore/normalize.median). Protein intensities were normalized according to the total intensity of each sample. *p* values between samples were obtained using a *t*-test. PCA analysis was generated in Bioladder (https://www.bioladder.cn/, accessed on 20 August 2022). Volcano plots and heatmaps were drawn in Bioinformatics (http://www.bioinformatics.com.cn/ accessed on 20 August 2022). GO and KEGG enrichment was conducted using Metascape (https://metascape.org/, accessed on 18 December 2021) and Kobas databases (http://kobas.cbi.pku.edu.cn/ accessed on 20 August 2022). 

### 2.6. Data Availability

The mass spectrometry raw data and the corresponding txt files have been deposited in the ProteomeXchange consortium (http://proteomecentral.proteomexchange.org, accessed on 14 October 2022) via the iProX partner repository [22] with a dataset identifier of IPX0003437002 [23].

### 2.7. ROS Detection

Cells were seeded in a 6-well plate and cultured overnight. DCFH-DA (S0033S, Beyotime, Shanghai, China) was diluted using DMEM medium without serum at the ratio of 1:1000. Cells were incubated with the diluent of DCFH-DA for 40 min at 37 °C followed by being washed 3 times using a DMEM medium. Fluorescence images were captured to evaluate ROS level.

### 2.8. Determination of Extra-Cellular Acidification Rate

The glycolysis level was reflected by the extra-cellular acidification rate. Cells were cultured in a seahorse plate (102601-100, Agilent Seahorse Technologies, Palo Alto, CA, USA) overnight with a density of 12,000 cells per well. The medium was a substitute for XF base medium (103334-100, Agilent Seahorse Technologies) supplement with 2 mM glutamine, 1 mM pyruvate and 10 mM glucose and then the plate was incubated for 1 h at 37 °C without CO_2_. XFe cell glycolysis stress test kit (103020-100, Agilent Seahorse Technologies) was used for ECAR detection. Results were analyzed using the software Wave 2.6.3 (Toronto, ON, Canada).

## 3. Results

### 3.1. Identification of Different Expression Proteins (DEPs)

Two LHON patients bearing m.G11778A and two healthy control subjects were included in the study. The skin tissues of 2 mm^3^ were excised from participants to obtain dispersed fibroblasts. We divided the samples into four groups (NC1, NC2, LHON1 and LHON2, each set consisted of three replicates) and performed a proteomic profile. The procedure is shown in Figure 1A. Firstly, to evaluate the reliability of the data, total protein differences between groups were assessed by principal component analysis (PCA). The results showed that samples were well separated by NC vs. LHON (PC1, 31.5%), indicating the feasibility of the data (Figure 1B). The expression of gross proteins was visualized on a heatmap plot using normalized log_2_ intensity values. As a result, a total of 2240 proteins were identified, and 860 proteins were significantly changed compared with the control cells. Of these differential expression proteins, 624 proteins were upregulated (Fold change ≥ 1.2, *p* ≤ 0.05), and 236 proteins were downregulated (Fold change ≤ 1.2, *p* ≤ 0.05) (Figure 1C), revealing favorable distinctiveness.

### 3.2. Annotation and Classification of Upregulated Proteins

To investigate the function of upregulated proteins of DEPs, GO (gene ontology) annotation was performed using the Metascape database (https://metascape.org/gp/index.Html#/main/step1, accessed on 13 July 2022) [24]. DEPs were classified into three categories, including biological process, cellular component and molecular function. The results demonstrated that among the 18 GO items (enrichment ranking), ribonucleoprotein complex biogenesis (BP), ribonucleoprotein complex (CC) and cadherin binding (MF) showed the highest enrichment of the three categories, with counts of 65, 120 and 66 (Figure 2A). In addition, we performed a KEGG (Kyoto Encyclopedia of Genes and Genomes) analysis in the KOBAS database (http://kobas.cbi.pku.edu.cn/genelist/, accessed on 13 July 2022) to explore differential signaling pathways. Firstly, in order to observe the types of cell activities affected by differential pathways intuitively, we drew a KEGG overview according to the classification in the KEGG PATHWAY Database (https://www.kegg.jp/kegg/pathway.html, accessed on 18 July 2022). Among the six categories of KEGG types, most of the differential pathways were classified to human diseases, followed by metabolic and organizational systems, and finally genetic information processing (Figure 2B). Notably, the metabolic pathway in the metabolism type revealed the highest enrichment, indicating that metabolism dysfunction may occur in LHON fibroblasts (m.G11778A). Specifically, the top 24 KEGG items were presented in Figure 2C. Results demonstrated that metabolic pathways (*n* = 85), ribosome (*n* = 49) and splicesome (*n* = 25) represented the three largest groups. Multiple metabolic pathways were identified, including carbon metabolism (*n* = 19), biosythesis of amino acids (*n* = 16), cysteine and methionine metabolism (*n* = 15), purine metabolism (*n* = 14), glycolysis or gluconeogenesis (*n* = 13), pyrimidine metabolism (*n* = 10) and drug metabolism-other enzyme (*n* = 10), which indicated that a variety of metabolic pathways might be affected in LHON fibroblasts (m.G11778A). 

### 3.3. Glycolysis Was Upregulated in LHON Fibroblasts (m.G11778A)

Aerobic respiration and anaerobic glycolysis are two major metabolic pathways that provide energy in cells [25]. In LHON fibroblasts, aerobic respiratory dysfunction caused by mtDNA mutations may affect anaerobic glycolysis. KEGG analysis showed that 13 DEPs enriched in the glycolysis or gluconeogenesis pathway, specifically PKM, ENO1, TPI1, LDHA, GAPDH, PGK1, PFKP, LDHB, PGAM1, GPI, PFKM and ENO2. The expression increased in LHON fibroblasts (m.G11778A) (Figure 3A). Further analysis indicated that the differential proteins above were positioned in the key catalytic locations, suggesting increased glycolysis (Figure 3B). Glycolysis is the process by which glucose metabolized to pyruvate under anaerobic conditions, during which H+ is produced and excreted into the extracellular medium, resulting in the acidification of the medium. Therefore, the ECAR (extra-cellular acidification rate) detection by seahorse reflected the level of glycolysis [26,27]. To elucidate changes in glycolysis at the cellular level, control and LHON fibroblasts were subjected to a seahorse glycolysis stress assay (Figure 3C). When cells were treated with glucose after incubating in the solution without glucose and sodium pyruvate, pyruvate was produced through the glycolysis pathway, and H+ was released at the same time. ECAR value at the time reflected the level of glycolysis under basal conditions (Figure 3D). Subsequently, the addition of oligomycin inhibited ATP synthase, and the energy production pathway switched to glycolysis. At the time, the ECAR value reflected the maximum glycolytic capacity of the cells (Figure 3E). Comprehensively, both basal and maximal ECAR values of LHON fibroblasts (m.G11778A) were higher than the control cells, indicating an enhanced level of glycolysis. 

In addition to participating in aerobic glycolysis, the 13 proteins are also involved in the cori cycle, gluconeogenesis, HIF-1 signaling pathway, carbon metabolism, etc. (Figure 3F). Together, these changes may lead to abnormalities in the metabolic network in LHON fibroblasts (m.G11778A).

### 3.4. Pathway Interaction Network Analysis of Upregulated Proteins

In order to better reveal the relationship between pathways involved in upregulated proteins, we drew an interaction network with the Metascape database. A total of 110 proteins (*p*-value ranks) were included. As shown in Figure 4, glycolysis and gluconeogenesis had a high enrichment. All pathways were closely connected with each other and enriched in adaptive immune system, cysteine and methionine metabolism, TP53 regulates metabolic genes, positive regulation of mRNA process, etc. The physiological interactions between these activities may contribute to metabolic disorders and apoptosis of optic nerve cells in LHON patients (m.G11778A).

### 3.5. Oxidoreductase Activity Was Downregulated in LHON Fibroblasts (m.G11778A)

Next, we performed GO annotations on the downregulated DEPs. The results showed that the lytic vacuole (CC, *n* = 37), intracellular protein transport (BP, *n* = 28) and oxidoreductase activity (MF, *n* = 27) had high enrichments (Figure 5A). In addition to energy metabolism regulation, mitochondrion also participated in redox reactions [25,28]. Thus, mitochondria dysfunction in LHON fibroblasts may lead to reduced oxidoreductase activity. The proteins enriched in oxidoreductase activity were listed in Figure 5B, and the expression was all significantly downregulated. As ROS is a product of dysregulated redox reactions, we detected the ROS level to clarify the changes in redox reactions. The results showed higher ROS in LHON fibroblasts compared to the control cells, indicating dysregulation of redox reactions (Figure 5C). The results were consistent with previous studies [29,30,31] and manifested the reliability of the data further.

### 3.6. Fatty Acid Metabolism Was Dysfunction in LHON Fibroblasts (m.G11778A)

To investigate the signal pathways affected by the downregulated DEPs, we created a KEGG overview map to investigate the affected cellular activities. The results indicated that more pathways were assigned to human diseases, followed by organic systems and metabolism (Figure 6A). Interestingly, the metabolic pathway was still the most enriched item among differential pathways (Figure 6A,B). The expression of enriched DEPs is presented in Figure 6C. Subsequently, we analyzed the specific metabolic activities involved in the DEPs and found that they mainly affected fatty acid metabolism, fatty acid degradation, tryptophan metabolism, thermogenesis, and oxidative phosphorylation (Figure 6D). These data demonstrated that fatty acid degradation was aberrant in LHON fibroblasts, which may lead to the accumulation of fatty acids. Although it is unclear whether superfluous fatty acids had an effect on RGC cells, accumulated fatty acids might cause lipotoxicity and inhibit cell survival.

### 3.7. Pathway Interaction Network Analysis of Downregulated Proteins

We mapped the interaction network for the downregulated proteins in Figure 7. The results showed that DEPs were strongly enriched in the glycosyl compound metabolic process, fatty acid metabolic process, hydrogen peroxide catabolic process etc., and these pathways were closely related to each other.

## 4. Discussion

Leber’s hereditary optic neuropathy is the most common and blindness-causing genetic eye disease for which there is still no effective treatment. Mitochondrial dysfunction caused by mtDNA mutations is a leading cause of the disease. Concurrently, mitochondrial damage also result in a variety of changes of cellular activities such as apoptosis [32,33], autophagy [34], metabolism [35,36], senescence [37,38], ion regulation [39,40] and signal transduction [41,42], which seriously affect cell survival. These mitochondrial damage-derived changes may serve as important secondary incidents that aggravate phenotypes or accelerate progression. Therefore, the study of molecular changes by proteomics is valuable to mining the affected primary and secondary incidents in LHON and is also conducive to a profound understanding of pathological changes.

Because of their consistent genotype and easy access, skin fibroblasts derived from patients have been used recently as cell models in genetic diseases research, although they may not be the specific diseased tissues [43,44]. Multiple studies indicated the derived fibroblasts as a valuable strategy for studying mitochondrial damage in neurological disorders, such as AD (Alzheimer’s Disease) [45,46], PD (Parkinson’s disease) [47,48], HD (Huntington’s disease) [49,50] and ALS (amyotrophic laternal sclerosis) [51,52]. Our previous studies also found mitochondrial dysfunction, respiration disorder and decreased cell viability in patients’ fibroblasts with m.G11778A [16]. Thus, to some extent, fibroblasts can stimulate the pathological process and can be used as an alternative model to explore molecular changes. In the current study, we performed proteomic profiling of two LHON fibroblasts with m.G11778A and control cells. A total of 860 differential expression proteins were identified, including 624 upregulated proteins and 236 downregulated proteins. In addition, bioinformatics analysis and molecular biology verification were conducted.

Firstly, a KEGG analysis of the upregulated proteins was performed. We showed that the metabolic pathway obtained the highest enrichment, indicating metabolism dysfunction in LHON fibroblasts. Among differential metabolic pathways, the glycolysis or gluconeogenesis pathway drew our attention. Aerobic respiration and anaerobic glycolysis are crucial pathways that provide energy to cells [53]. Under normal circumstances, cells obtain a large amount of ATP through aerobic respiration with glycolysis inhibition. But the mutation of mtDNA reduces the level of respiration, and cells may produce ATP as much as possible by enhancing glycolysis to provide sufficient energy for signaling. A recent study found that neurons activated rapid and maximized aerobic glycolysis in astrocytes for energy to prevent shortages in ATP production during information processing, which suggested that metabolic reprogramming presented in the nervous system under pathological conditions and might also occur in RGCs [54]. We did detect elevated glycolysis in LHON fibroblasts. The expression of catalytic enzymes guarded at pivotal locations was upregulated, suggesting an increase in glycolysis level. However, this metabolic plasticity, especially the conversion of aerobic respiration to glycolysis, might lead to an increased level of cellular acidification while maximizing energy supply, which in turn accelerates apoptosis. Unfortunately, RGC cells might not have strategies to ameliorate this adverse environment like the tumor cells. Therefore, an increased energy supply combined with inhibition of glycolysis might contribute to delaying disease progression. The upregulated proteins in LHON fibroblasts not only led to metabolism dysfunction but also participated in the regulation of the adaptive immune system, axon guidance, TP53 regulates metabolic genes and other processes. These changes are closely associated with and organized the intricate network of cellular activities.

Next, we performed a bioinformatic analysis of the downregulated proteins. GO results revealed a high enrichment in oxidoreductase activity. This was plausible because the respiratory chain damage caused by mtDNA mutations resulted in the dysregulation of electron balance, decreased membrane potential and the inhibition of oxidoreductase activity, which placed the cells under a condition of further stress. As a result, ROS accumulation and protein oxidation aggravated mitochondrial damage [55]. Our results also showed elevated ROS levels in LHON fibroblasts. Besides, previous studies also found that ROS has increased in LHON heterozygous cells [56] and the retina of mt-Nd6 mutant mice [30], which were consistent with our omics results. Interestingly, KEGG analysis of the downregulated DEPs showed that the metabolic pathway still had the highest enrichment. Subsequent analysis indicated that these proteins were mainly involved in fatty acid metabolism, fatty acid degradation and oxidative phosphorylation, suggesting the accumulation of fatty acids. Activation, transfer and β-oxidation are the three stages of fatty acid degradation. β-oxidation occurs in the mitochondrial matrix, and mitochondria disorder caused by mtDNA variation may inhibit the β-oxidation process, which in turn leads to metabolism dysregulation of fatty acid. An early study found that respiratory chain complex I inhibitor altered fatty acid biosynthesis and β-oxidation in neuroblastoma cells [57]. Additionally, Leong et al. also reported that β-oxidation was inhibited in the presence of respiratory chain complex I mutations [58]. Morvan and Demidem found complex I mutation increased intracellular fatty acids level using NMR metabolomics [44]. These studies were consistent with our results and indicated fatty acid degradation was presented in LHON fibroblasts. Although fatty acids act as “energy vectors”, excess fatty acids may aggravate mitochondrial damage, such as an increase in ROS and mitochondrial proton conductance (uncoupling). The lipotoxicity may further accelerate the apoptosis of RGCs [59,60].

In conclusion, we made a proteomic profile of skin-derived fibroblasts of LHON patients (m.G11778A) and found increased glycolysis, decreased oxidoreductase activity, and dysregulation of fatty acid metabolism (Figure 8). These changes might represent the concurrent incidents in the cells during the LHON progression, and strategies targeting these pathways might be helpful for developing novel treatments.

## Figures and Tables

**Figure 1 biomolecules-12-01568-f001:**
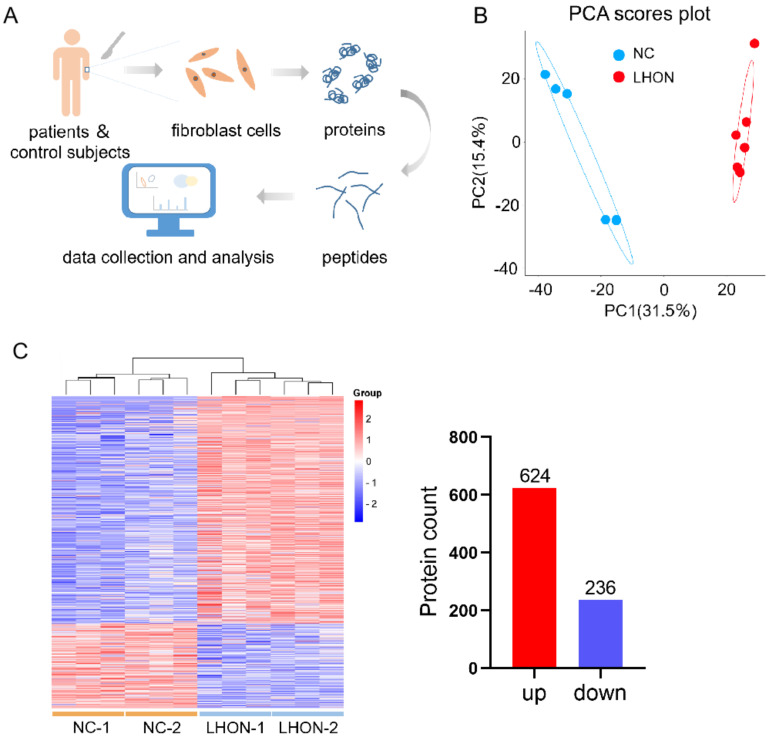
Proteome overview of LHON fibroblasts (m.G11778A). (**A**) The strategy of this study. Scattered fibroblasts were isolated from skin tissue obtained from LHON patients bearing m.G11778A and control subjects. Peptides were extracted after cell lysis, and then proteomic and bioinformatics analyses were performed. (**B**) PCA of total proteins between fibroblasts of LHON and control. (**C**) Heatmap plot showing DEPs (differential expression proteins) between fibroblasts of LHON and control. A total of 860 DEPs were identified, including 624 upregulated proteins (FC ≥ 1.2, *p* ≤ 0.05) and 236 downregulated proteins (FC ≤ 1.2, *p* ≤ 0.05).

**Figure 2 biomolecules-12-01568-f002:**
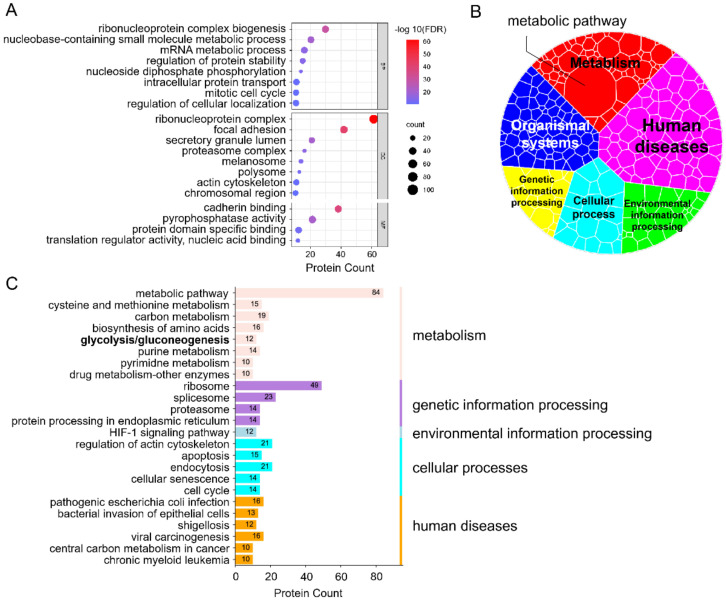
GO and KEGG analysis of upregulated proteins. (**A**) GO enrichment of the upregulated DEPs using Metascape database, including BP (biological process), CC (cellular component) and MF (molecular function). The plot listed top items based on –log 10(FDR) rank. (**B**) Voronoi tree map showed attributions of differential pathways according to KEGG database. Each padding represents a differential pathway, and filled areas represent enrichment count. (**C**) KEGG enrichment of DEPs using kobas database showed top items ranked by count.

**Figure 3 biomolecules-12-01568-f003:**
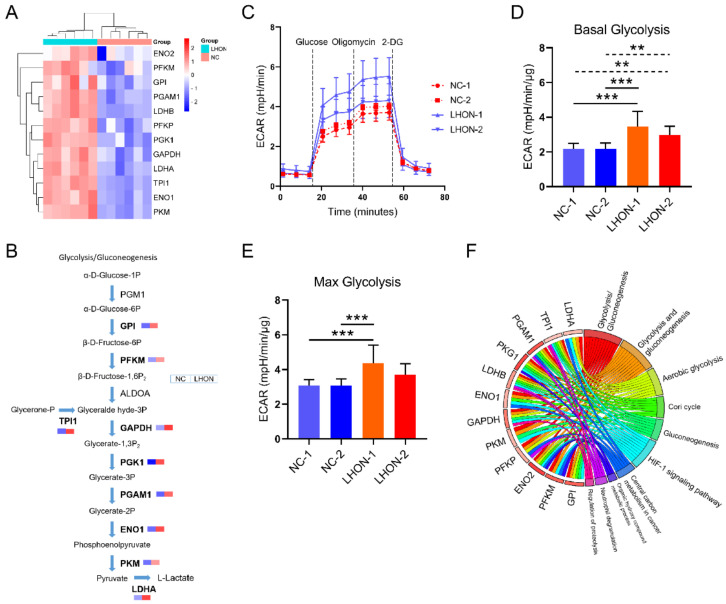
Increased glycolysis level in LHON fibroblasts (m.G11778A). (**A**) Clustered heatmap displaying expression of DEPs participated in glycolysis. (**B**) Flowchart showing glycolysis process. DEPs are emphasized in bold, and the color scale indicates the expression level. (**C**) ECAR (extra-cellular acidification rate) of fibroblasts was measured using XFe96 seahorse analysis. The basal (**D**) and max (**E**) ECAR values were calculated. ** *p* ≤ 0.05, *** *p* ≤ 0.01. (**F**) GO chord plot showed different pathways involved in glycolysis DEPs.

**Figure 4 biomolecules-12-01568-f004:**
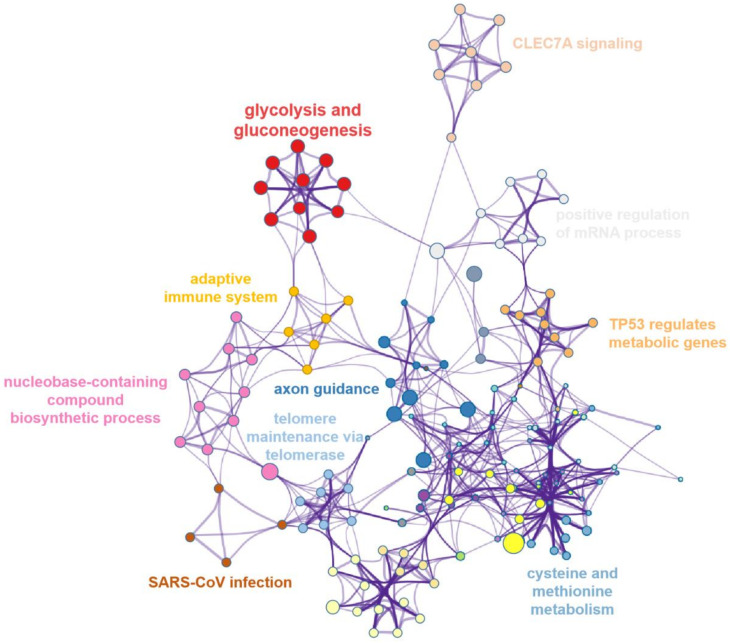
Network of enriched items of the upregulated DEPs. Terms are colored by cluster ID.

**Figure 5 biomolecules-12-01568-f005:**
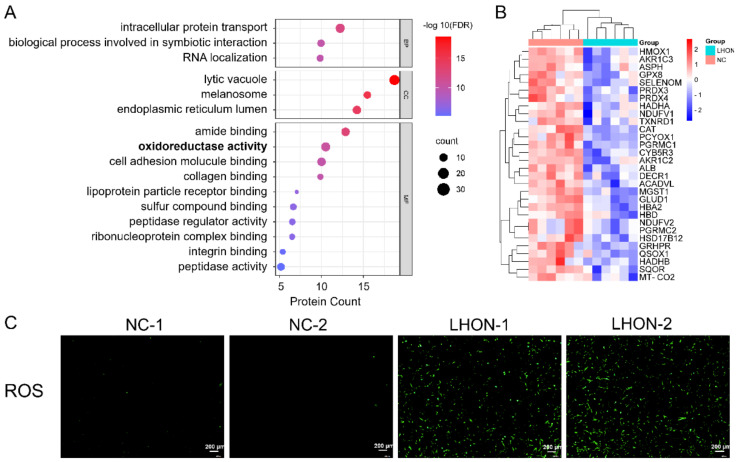
Downregulated oxidoreductase activity in LHON fibroblasts (m.G11778A). (**A**) Metascape enrichment of the downregulated DEPs was plotted. BP: biological process, CC: cellular component, MF: molecular function. Items were ranked by –log 10(FDR). (**B**) Heatmap showed downregulated differential proteins participated in oxidoreductase activity. (**C**) ROS level was detected in LHON and control fibroblasts. Scale bars represent 200 μm.

**Figure 6 biomolecules-12-01568-f006:**
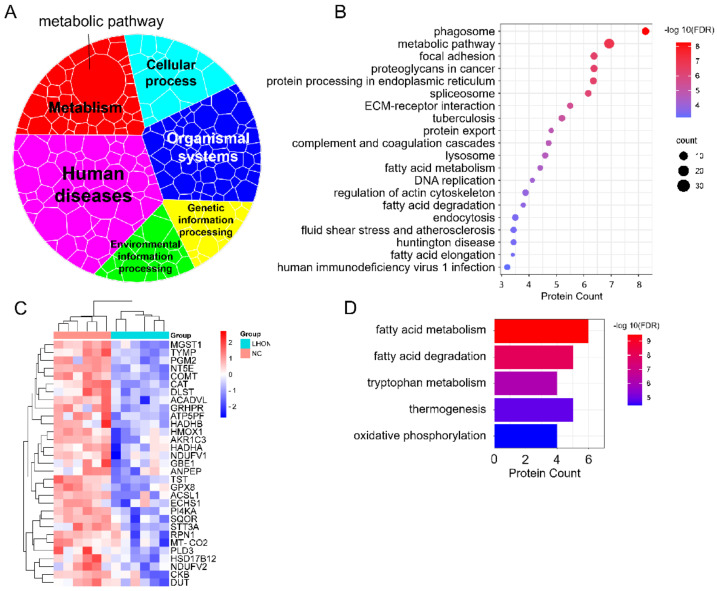
Fatty acid metabolism dysfunction in LHON fibroblasts (m.G11778A). (**A**) Voronoi tree map of differential pathways enriched by downregulated DEPs. (**B**) Kobas KEGG enrichment of downregulated DEPs. The top 20 items ranked by –log 10(FDR) are listed. (**C**) Downregulated DEPs involved in metabolic pathways were plotted as a heatmap. (**D**) Detailed KEGG enrichment of the downregulated DEPs participated in metabolic pathways. Items were ranked by –log 10(FDR).

**Figure 7 biomolecules-12-01568-f007:**
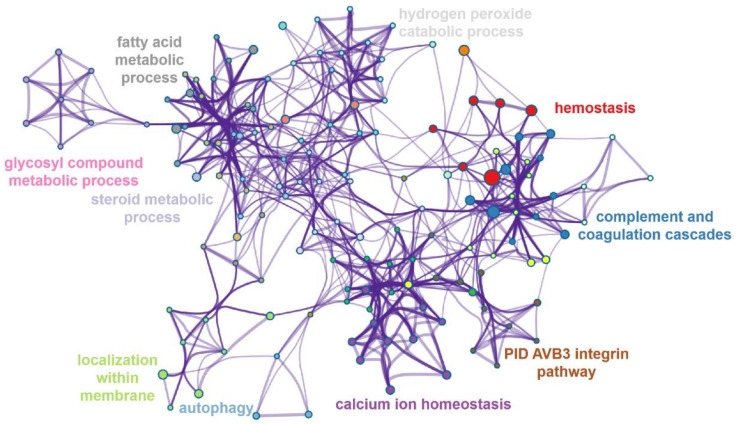
Network of enriched items of the downregulated DEPs. Terms were colored by cluster-ID.

**Figure 8 biomolecules-12-01568-f008:**
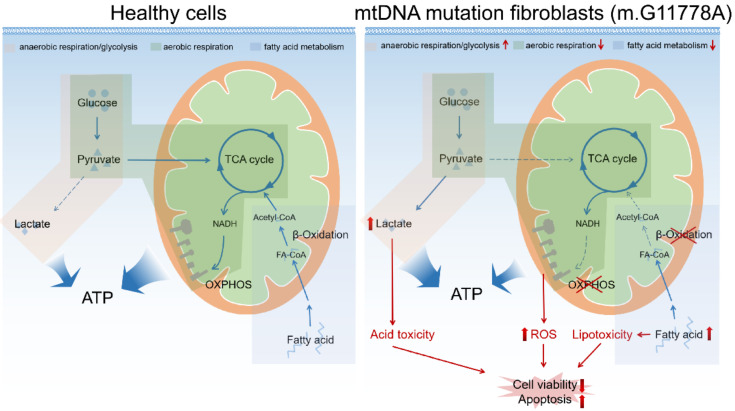
Graphical summary of metabolism changes in LHON fibroblasts bearing m.G11778A.

## Data Availability

The mass spectrometry raw data and the corresponding txt files have been deposited in the ProteomeXchange consortium (http://proteomecentral.proteomexchange.org, accessed on 14 October 2022) via the iProX partner repository with a dataset identifier of IPX0003437002.

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
