# Peer review of "Proteomic Profiling Reveals Increased Glycolysis, Decreased Oxidoreductase Activity and Fatty Acid Degradation in Skin Derived Fibroblasts from LHON Patients Bearing m.G11778A"

_biomolecules, 2022, doi:10.3390/biom12111568_

Round 1
Reviewer 1 Report
This is an interesting study by the authors that shed light on the molecular changes which can be used for the development of therapy.
Author Response
Re: Thanks very much for your review.
Reviewer 2 Report
This manuscript described a proteomics application on LHON fibroblast study. The data analysis and figures are well presented. Overall, the article is certainly of interest to researchers and deserves publication in Biomolecules. But I have a few points which should be addressed before publication.
1. What is the difference between this work with the reference 17, Tun A. et al. and what is the protein quality control system provides in this manuscript?
2. In line 338, the trap column "10×0.3", what is 10 mean?
3. What is the length of trapping and analytical column?
4. What is the gradient and the total gradient time?
5. Mass spectrometry setting is not clear. What is the MS resolution of DDA and DIA setting? What is the isolation window for MS/MS? What is the collision energy? For DIA, what is the M/Z isolation windows? what is MS/MS spectra resolution?
6. For proteomics data, please put all the raw data in the public data repository. http://www.proteomexchange.org/ or https://www.ebi.ac.uk/pride/archive/ or https://repository.jpostdb.org/ anyone will be working.
7. How many peptide amount were injected into column?
http://www.proteomexchange.org
Author Response
Re: Thanks very much for your review. We have received your questions and made reply. Please feel free to inform us if there is any need for further modification, thank you.
- What is the difference between this work with the reference 17, Tun A. et al. and what is the protein quality control system provides in this manuscript?
Re: Thanks very much for your attention. Our work is different from reference 17:
- Firstly, in terms of methodology, Tun A. et al. extracted mitochondrial proteins for proteomic analysis. We obtained the whole proteins for proteomic detection.
- Secondly, in terms of objective, Tun A. et al. investigated the difference of mitochondrial proteomics between affected and unaffected LHON patients bearing m.G11778A to answer the causes of phenotypic heterogeneity under the same pathogenic variation. We performed proteomic investigation after extracting whole proteins and analyzed the different between control and affected LHON patients with m.G11778A to reveal pathogenicity mechanisms.
- Finally, in terms of results, Tun A. et al. found decreased OXPHOS, insufficient bioenergetics and dysfunction of mitochondrial protein in LHON fibroblasts. It is interesting to observe the similarities in mitochondrial protein expression profiles between affected and unaffected m.G11778A carriers, which suggest pathways other than mitochondria may be involved in the regulation of heterogeneity. In our work, we also found decreased oxidoreductase activity in LHON fibroblasts which was consistent with Tun A. et al. Besides, we reported increased glycolysis and dysfunction of fatty acid metabolism in LHON and these data represented in-depth mechanisms of mitochondrial dysfunction mediated by the mutation.
In this work, Pearson correlation analysis was performed among the samples to evaluate quality. The results were showed in the follow picture (please find attachment). The correlation coefficients among samples within and between groups reached 0.72 and above, showing the high repeatability of the data (NC-1 in manuscript is CG in the following picture. NC-2: CS, LHON-1: LW, LHON-2: LM.).
- In line 338, the trap column "10×0.3", what is 10 mean?
Re: Thanks for your question. In this description, 10 means 10 mm, which is the length of trap column. We have optimized this description in revised manuscript.
Revised position: line 345-347: The digested peptides (1.5 μg) were loaded into the Trap column of AB SCIEX (10 mm ×0.3 mm, C18 packing size was 5 μm, 120 A) with phase A (0.1% formic acid, 2% ACN, 97.9% water), and the flow rate was 10 μl/ min.
- What is the length of trapping and analytical column?
Re: Thanks for your question. The length of trapping column is 10 mm and the length of analytical column is 150 mm. We have modified this description in revised manuscript.
Revised position: line 346: 10 mm ×0.3 mm; line 350: 150 mm×0.3mm.
- What is the gradient and the total gradient time?
Re: Thanks for your review. In this study, different gradients of phase B (97.9% acetonitrile, 2% water, 0.1% formic acid, v/v/v) was used to elute the Trap column for 91 min at a flow rate of 400 nL/ min. We have added this description in revised manuscript.
Revised position: line 347-351: Different gradients of phase B (97.9% acetonitrile, 2% water, 0.1% formic acid, v/v/v) was used to elute the Trap column for 91 min at a flow rate of 400 nL/ min.
- Mass spectrometry setting is not clear. What is the MS resolutionof DDA and DIA setting? What is the isolation window for MS/MS? What is the collision energy? For DIA, what is the M/Z isolation windows? what is MS/MS spectra resolution?
Re: Thanks for your attention.
- The mass spectrometer we used was Triple TOF 6600 (AB SCIEX, Framingham, MA, USA). This instrument uses the default resolution for data acquisition without resetting. So we have no record. If this parameter should be mentioned in the paper, please let us know and we will recheck the raw setting data.
- For isolation window, TOF 6600 has different parameters with equipment from Thermo Fisher. Briefly, in TOF 6600, Q1 was unit resolution and width was 0.7.
- Collision energy (CE) was not constant. It changes according to the formula: ∣CE∣=(slope)*(m/z)+(intercept). The following figure shows the setting interface (please find attachment).
- For DIA analysis, a variable window assay calculator (AB Sciex, version 1.1, Framingham, MA, USA) was used to optimize the 100 scanning windows. The accumulation time for MS1 was 50 ms, and 40 ms for MS2. The ion spray floating voltage was 2300 V.We have added these modifications in the revised manuscript.
Revised position: line 356-358: For DIA analysis, a variable window assay calculator (AB Sciex, version 1.1, Framingham, MA, USA) was used to optimize the 100 scanning windows. The accumulation time for MS1 was 50 ms, and 40 ms for MS2. The ion spray floating voltage was 2300 V.
- We are sorry for no record of spectra resolution also because of no required setup in Triple TOF 6600. If this parameter was needed, we will recheck the raw setting data.
- For proteomics data, please put all the raw data in the public data repository. http://www.proteomexchange.org/ or https://www.ebi.ac.uk/pride/archive/ or https://repository.jpostdb.org/ anyone will be working.
Re: Thanks for your suggestion. The mass spectrometry raw data and the corresponding txt files have been deposited in the ProteomeXchange consortium (http://proteomecentral.proteomexchange.org) via the iProX partner repository with a dataset identifier of IPX0003437002. We have added this in Materials and Methods of revised manuscript.
Revised position: line 373-375:The mass spectrometry raw data and the corresponding txt files have been deposited in the Proteome Xchange consortium (http://proteomecentral.proteomexchange.org) via the iProX partner repository [59] with a dataset identifier of PXD028202 [60].
References:
- Ma, J.; Chen, T.; Wu, S.; Yang, C.; Bai, M.; Shu, K.; Li, K.; Zhang, G.; Jin, Z.; He, F.; Hermjakob, H.; Zhu, Y. iProX: an in-tegrated proteome resource. Nucleic acids research 2019, 47, D1211–D1217, doi: 10.1093/nar/gky869.
- Jin, X.; Liu, J.; Wang, W.; Li, J.; Liu, G.; Qiu, R.; Yang, M.; Liu, M.; Yang, L.; Du, X.; Lei, B. Identification of age-associated proteins and functional alterations in human retinal pigment epithelium. Genomics, proteomics & bioinformatics 2022, S1672-0229(22)00077-8, Advance online publication, doi:10.1016/j.gpb.2022.06.001.
- How many peptide amount were injected into column?
Re: A total of 1.5 μg peptide were injected into column. We have modified this description in the revised manuscript.
Revised position: line 345-347: The digested peptides (1.5 μg) were loaded into the Trap column of AB SCIEX (10 mm ×0.3 mm, C18 packing size was 5 μm, 120 A) with phase A (0.1% formic acid, 2% ACN, 97.9% water), and the flow rate was 10 μl/ min.

Reviewer 3 Report
In this manuscript by Lei and colleagues, the authors utilized the systems biology approach and characterized the proteomic changes caused by the most common mitochondrial G11778A mutation in the fibroblasts derived from LHON patients and healthy donors. The authors uncovered enriched pathways of increased glycolysis, decreased oxidoreductase activity and decreased fatty acid metabolism in the LHON fibroblasts. Importantly, the authors further validated their bioinformatic findings with functional analyses. They found increased extra-cellular acidification rate for glycolysis stress and dramatically increased ROS levels in LHON fibroblast. Likewise, their findings on fatty acid metabolism are supported by another study. Overall, this study provides a novel proteomic dataset, which will serve as a useful resource for studying cellular changes in LHON. Also, bioinformatic analyses and experimental validations were well-executed. The following comments are meant to provide constructive feedback to further strengthen this study.
(1) Given LHON has a distinct sex difference, it is important to document the sex information of donors. It would be also helpful to include additional metadata of donors, such as age and medical history.
(2) In line 24, it is unclear what “decreased levels of redox activity” means. Does it mean decreased reduction activity, or decreased oxidation activity?
(3) In line 33, it should be “which may lead”.
(4) It is not clear what n means in lines 112-117? Does it indicate the number of proteins identified in the pathway?
(5) The titles of x axis in Fig.2A and Fig.5A should be labelled. It should also be mentioned how pathways were ranked, by fold change or by -log(p-value/FDR)
(6) It would be recommended using FDR instead of p-value for pathway enrichment analysis.
Author Response
Re: Thanks very much for your review. We have received your questions and made reply. Please feel free to inform us if there is any need for further modification, thank you. Attached please find responses with revised figures.
- Given LHON has a distinct sex difference, it is important to document the sex information of donors. It would be also helpful to include additional metadata of donors, such as age and medical history.
Re:Thank you for your suggestion. The four participants came from 4 families, all male and aged between 20 and 32. The detailed information of LHON patients (LHON-1 and LHON-2) was described in the previous article [Yao et al., 2022] and the two controls did not have ophthalmic diseases. We have added more detailed description in revised manuscript.
Revised potion: line311-318: Male participants came from 4 families. Two controls were around 20 years old. Ophthalmological examinations showed the binocular visual acuity in both eyes was 1.0 or above, and noabnormality was found in visual field examination. The patients LHON-1 and LHON-2 were 28 and 32 years old. The ophthalmic examinations showed pathological optic discs with incomplete boundaries and the optic nerve has atrophy of varying degrees. Gene sequencing reported they both suffered mutation m.G11778A. Finally they were diagnosed as LHON bearing m.G11778A. Detailed information could be found in a previous article [16].
Reference:
[16] Yao, S.; Zhou, Q.; Yang, M.; Li, Y.; Jin, X.; Guo, Q.; Yang, L.; Qin, F.; Lei, B. Multi-mtDNA Variants May Be a Factor Con-tributing to Mitochondrial Function Variety in the Skin-Derived Fibroblasts of Leber's Hereditary Optic Neuropathy Pa-tients. Frontiers in Molecular Neuroscience 2022, 15, doi:10.3389/fnmol.2022.920221.
- In line 24, it is unclear what “decreased levels of redox activity” means. Does it mean decreased reduction activity, or decreased oxidation activity?
Re: Thanks for your review. We agree with you that this word was unclear. In the study, we found many oxidoreductases were downregulated and the level of ROS was increased in LHON fibroblasts. So the description of “decreased oxidoreductase activity” may be more appropriate. We have corrected this word in revised manuscript.
Revised position: line 23-25: Increased levels of glycolysis, decreased oxidoreductase activity and fatty acid metabolism could represent the in-depth mechanisms of mitochondrial dysfunction mediated by the mutation.
- In line 33, it should be “which may lead”.
Re: Thanks for your correction. We have corrected this mistake.
Revised position: lines 32-34: The average onset of patients is around 20 years old, with subacute vision loss in both eyes, which may lead to blindness and bring a heavy burden to individuals and their families
- It is not clear what n means in lines 112-117? Does it indicate the number of proteins identified in the pathway?
Re: Thanks for your question. You are correct. The n means the number of proteins identified in the pathway.
- The titles of x axis in Fig.2A and Fig. 5A should be labelled. It should also be mentioned how pathways were ranked, by fold change or by -log(p-value/FDR)
Re: Thanks for your attention. We have corrected these figures and added description in revised manuscript.
Revised positon:
Line119-123:
Fig. 2A
Figure 2. GO and KEGG analysis of upregulated proteins.
- GO enrichment of the upregulated DEPs using metascape database including BP (biological process), CC (cellular component) and MF (molecular function). The plot listed top items based on –log 10(FDR) rank.
Line188-192:
Fig. 5A
Figure 5. Downregulated oxidoreductase activity in LHON fibroblasts (m.G11778A).
(A) Metascape enrichment of the downregulated DEPs was plotted. BP: biological process, CC: cellular component, MF: molecular function. Items were ranked by –log 10(FDR).
- It would be recommended using FDR instead of p-value for pathway enrichment analysis.
Re: Thanks for your suggestion. We analyzed again using FDR according to your recommend. The results showed no changes in order. We have revised the figures and legends in the revised manuscript.
Revised position:
Figure 2. GO and KEGG analysis of upregulated proteins.
(A) GO enrichment of the upregulated DEPs using metascape database including BP (biological process), CC (cellular component) and MF (molecular function). The plot listed top items based on –log 10(FDR) rank. (B) Voronoitreemap showed attributions of differential pathways according to KEGG database. Each padding represented a dif-ferential pathway and filled area represented enrichment count. (C) KEGG enrichment of DEPs using kobas database showed top items ranked by count.
(comment: In Fig. 2C, all upregulated proteins were performed KEGG pathway analysis and items were ranked by –log 10(FDR) firstly. Then the top 25 items were classified into 5 groups according to KEGG database and ranked by count. )
Figure 5. Downregulated oxidoreductase activity in LHON fibroblasts (m.G11778A).
(A) Metascape enrichment of the downregulated DEPs was plotted. BP: biological process, CC: cellular component, MF: molecular function. Items were ranked by –log 10(FDR). (B) Heatmap showed downregulated differential proteins participated in ox-idoreductase activity. (C) ROS level was detected in LHON and control fibroblasts.
Figure 6. Fatty acid metabolism dysfunction in LHON fibroblasts (m.G11778A).
(A) Voronoitreemap of differential pathways enriched by downregulated DEPs. (B) Kobas KEGG enrichment of downregulated DEPs. A total of top 20 items ranked by –log 10(FDR) was listed. (C) Downregulated DEPs involved in metabolic pathway were plotted as a heatmap. (D) Detailed KEGG enrichment of the downregulated DEPs par-ticipated in metabolic pathway. Items were ranked by –log 10(FDR).

Reviewer 4 Report
LHON is an important blinding inherited optic neuropathy. In this study, the authors used the skin fibroblasts derived from LHON patients and healthy objects and conducted the proteomic analysis to identify the potential proteins, signaling pathways and cellular activities involved in LHON. This study would provide novel insights into the pathogenesis of LHON disease and also provide a novel model for studying LHON disease. Overall, the paper is well-written and the study is well designed. There are some details that need to be provided prior to publication.
(1) Pay attention to grammar for word Superscript, such as A volume of 2 mm3 of skin tissue and word subscript NH4HCO3, H2O2, et ac
(2) Provide more details for some experiments, for example, the wavelength for ROS detection. The websites for bioinformatic analysis such as UniProt Swiss human database
(3) Glycolysis and fatty acid metabolism are tightly associated with the pathogenesis of human diseases especially for the mitochondrial disease. The authors can discuss the implication of the dysfunction of fatty acid metabolism in LHON disease.
(4) Whether the key proteins involved in tricarboxylic acid cycle was found to be associated with LHON disease?
(5) Provided more details about LHON patients used for Fibroblasts isolation?
(6) Provide the scale bar for IF images.
Author Response
Re: Thanks very much for your review. We have received your questions and made reply. Please feel free to inform us if there is any need for further modification, thank you. Attached please find responses with revised figure.
- Pay attention to grammar for word Superscript, such as A volume of 2 mm3 of skin tissue and word subscript NH4HCO3, H2O2, et ac
Re: Thanks very much for your review. We have rechecked the whole paper again and modified some incorrect words with superscript.
- Provide more details for some experiments, for example, the wavelength for ROS detection. The websites for bioinformatic analysis such as UniProt Swiss human database
Re: Thanks for your attention. In the study, we used fluorescent probe (DCFH-DA) to detect ROS levels. DCFH-DA is a probe without fluorescence and can pass through the cell membrane freely. After entering the cytoplasm, it can be hydrolyzed by the esterase in the cell to generate DCFH. DCFH can be easily loaded into the cell because it cannot penetrate the cell membrane. And then reactive oxygen species (ROS) in cells can oxidize non fluorescent DCFH to generate fluorescent DCF. Thus, the fluorescence intensity of DCF can reflect the level of ROS.
The UniProt Swiss human database (UniProt release 2020_11) was used as a library database. We described these information in the revised manuscript.
Revised position: line361-362: The UniProt Swiss human database (UniProt release 2020_11) was used as a library database.
- Glycolysis and fatty acid metabolism are tightly associated with the pathogenesis of human diseases especially for the mitochondrial disease. The authors can discuss the implication of the dysfunction of fatty acid metabolism in LHON disease.
Re: Thanks for your suggestion. We agree with you that glycolysis and fatty acid metabolism are tightly associated with many human diseases. Fatty acid has a critical role in energy metabolism. β-oxidation and biosynthesis are two main biological processes in fatty acid metabolism, and malfunction of pathway can lead to diseases. Disorder of fatty acid metabolism is a characteristic in ophthalmic diseases such as retinopathy and dry eye. Mitochondrial dysfunction causes disorder of β-oxidation, resulting in fatty acid accumulation. This change, in turn, worsens mitochondrial damage and leading to ROS upregulation and mitochondrial uncoupling, which will be detrimental to cell health and also known as lipotoxicity. Morvan and Demidem found complex I mutation increased intracellular fatty acids level. We also observed dysfunction of fatty acid metabolism in fibroblasts bearing m.G11778A. These evidences reminded that the abnormal metabolism of fatty acid may participate in LHON. Besides, RGCs were extremely sensitive to mitochondrial damage, and the mitochondria load caused by fatty acid accumulation may accelerate cell death, thereby aggravating the phenotype of LHON. We mentioned this in the revised manuscript.
Revised position: line285-298: Subsequent analysis indicated that these proteins were mainly involved in fatty acid metabolism, fatty acid degradation and oxidative phosphorylation, suggesting the accumulation of fatty acids. Activation, transfer and β-oxidation are the 3 stages of fatty acid degradation. β-oxidation occurs in the mitochondrial matrix, and mitochondria disorder caused by mtDNA variation may inhibit the β-oxidation process, which in turn leads to metabolism dysregulation of fatty acid. An early study found that respiratory chain complex I inhibitor altered fatty acid biosynthesis and β-oxidation in neuroblastoma cells [54]. Additionally, Leong et al. also reported that β-oxidation was inhibited in the presence of respiratory chain complex I mutations [55]. Morvan and Demidem found complex I mutation increased intracellular fatty acids level using NMR metabolomics [41]. These studies were consistent with our results and indicated fatty acid degradation was presented in LHON fibroblasts. Although fatty acids act as “energy vectors”, excess fatty acids may aggravate mitochondrial damage, such as increase in ROS and mitochondrial proton conductance (uncoupling). The lipotoxicity may further accelerate apoptosis of RGCs [56,57].
References:
- Morvan, D.; Demidem, A. NMR metabolomics of fibroblasts with inherited mitochondrial Complex I mutation reveals treatment-reversible lipid and amino acid metabolism alterations. Metabolomics : Official journal of the Metabolomic Society 2018, 14, 55, doi:10.1007/s11306-018-1345-9.
- Worth, A.J.; Basu, S.S.; Snyder, N.W.; Mesaros, C.; Blair, I.A. Inhibition of neuronal cell mitochondrial complex I with rotenone increases lipid beta-oxidation, supporting acetyl-coenzyme A levels. The Journal of biological chemistry 2014, 289, 26895-26903, doi:10.1074/jbc.M114.591354.
- Leong, D.W.; Komen, J.C.; Hewitt, C.A.; Arnaud, E.; McKenzie, M.; Phipson, B.; Bahlo, M.; Laskowski, A.; Kinkel, S.A.; Davey, G.M.; et al. Proteomic and metabolomic analyses of mitochondrial complex I-deficient mouse model generated by spontaneous B2 short interspersed nuclear element (SINE) insertion into NADH dehydrogenase (ubiquinone) Fe-S protein 4 (Ndufs4) gene. The Journal of biological chemistry 2012, 287, 20652-20663, doi:10.1074/jbc.M111.327601.
- Schonfeld, P.; Wojtczak, L. Fatty acids as modulators of the cellular production of reactive oxygen species. Free radical biology & medicine 2008, 45, 231-241, doi:10.1016/j.freeradbiomed.2008.04.029.
- Rial, E.; Rodriguez-Sanchez, L.; Gallardo-Vara, E.; Zaragoza, P.; Moyano, E.; Gonzalez-Barroso, M.M. Lipotoxicity, fatty acid uncoupling and mitochondrial carrier function. Biochimica et biophysica acta 2010, 1797, 800-806, doi:10.1016/j.bbabio.2010.04.001.
- Whether the key proteins involved in tricarboxylic acid cyclewas found to be associated with LHON disease?
Re: Thanks for your question. At present, there is no direct evidence that the key proteins in TCA process related to LHON. But Tun A. et al. found some proteins involved in TCA, such as FAD-dependent succinate dehydrogenase, NAD-dependent 2-oxoglutarate dehydrogenase, pyruvate dehydrogenase E1component alpha subunit and dihydrolipoyl dehydrogenase weredownregulated in m.G11778A fibroblasts, which revealed the inhibition of TCA. In addition, the upregulated ROS level in m.G11778A fibroblasts was also unfavorable to TCA. However, in our results, the abnormal TCA process was not found. More experiments are needed to prove this point of view.
- Provided more details about LHON patients used for Fibroblasts isolation?
Re: Thanks for your question. The detailed information of LHON patients was described in our previous article [Yao et al., 2022]. Briefly, they were all male, around 30 years old and came from two Chinese families. Both had visual acuity below 0.15. The ophthalmic examinations showed pathological optic discs with incomplete boundaries and the optic nerve has atrophy of varying degrees. Gene sequencing reported they both suffered mutation m.G11778A. Finally they were diagnosed as LHON bearing m.G11778A. We have added more description in the revised manuscript.
Revised position: line311-318: Male participants came from 4 families. Two controls were around 20 years old. Ophthalmological examinations showed the binocular visual acuity in both eyes was 1.0 or above, and noabnormality was found in visual field examination. The patients LHON-1 and LHON-2 were 28 and 32 years old. The ophthalmic examinations showed pathological optic discs with incomplete boundaries and the optic nerve has atrophy of varying degrees. Gene sequencing reported they both suffered mutation m.G11778A. Finally they were diagnosed as LHON bearing m.G11778A. Detailed information could be found in a previous article [16].
References:
[16] Yao, S.; Zhou, Q.; Yang, M.; Li, Y.; Jin, X.; Guo, Q.; Yang, L.; Qin, F.; Lei, B. Multi-mtDNA Variants May Be a Factor Con-tributing to Mitochondrial Function Variety in the Skin-Derived Fibroblasts of Leber's Hereditary Optic Neuropathy Pa-tients. Frontiers in Molecular Neuroscience 2022, 15, doi:10.3389/fnmol.2022.920221.
- Provide the scale bar for IF images.
Re: Thanks for your suggestion. Actually, the scale bars existed in the lower right corner of the images, but they are too small. We redraw clear scale bars in the images as following.